# Left Ventricle Wall Motion Analysis with Real-Time MRI Feature Tracking in Heart Failure Patients: A Pilot Study

**DOI:** 10.3390/diagnostics12122946

**Published:** 2022-11-25

**Authors:** Yu (Yulee) Li, Jason Craft, Yang (Josh) Cheng, Kathleen Gliganic, William Schapiro, Jie (Jane) Cao

**Affiliations:** 1Cardiac Imaging, DeMatteis Center for Cardiac Research and Education, St. Fracis Hospital & Heart Center, Greenvale, NY 11548, USA; 2Biomedical Engineering, State University of New York at Stony Brook, Stony Brook, NY 11794, USA; 3Clinical Medicine, State University of New York at Stony Brook, Stony Brook, NY 11794, USA

**Keywords:** heart failure, real-time imaging, feature tracking, volumetric measurements, LV wall motion

## Abstract

Volumetric measurements with cardiac magnetic resonance imaging (MRI) are effective for evaluating heart failure (HF) with systolic dysfunction that typically induces a lower ejection fraction (EF) than normal (<50%) while they are not sensitive to diastolic dysfunction in HF patients with preserved EF (≥50%). This work is to investigate whether HF evaluation with cardiac MRI can be improved with real-time MRI feature tracking. In a cardiac MRI study, we recruited 16 healthy volunteers, 8 HF patients with EF < 50% and 10 HF patients with preserved EF. Using real-time feature tracking, a cardiac MRI index, torsion correlation, was calculated which evaluated the correlation of torsional and radial wall motion in the left ventricle (LV) over a series of sequential cardiac cycles. The HF patients with preserved EF and the healthy volunteers presented significant difference in torsion correlation (one-way ANOVA, *p* < 0.001). In the scatter plots of EF against torsion correlation, the HF patients with EF < 50%, the HF patients with preserved EF and the healthy volunteers were well differentiated, indicating that real-time MRI feature tracking provided LV function assessment complementary to volumetric measurements. This study demonstrated the potential of cardiac MRI for evaluating both systolic and diastolic dysfunction in HF patients.

## 1. Introduction

Heart failure (HF) is a complex clinical syndrome due to a compromise of the ventricular ability to fill with and eject blood [1,2]. It may be caused by a number of widely different cardiovascular diseases, such as coronary artery disease, hypertension, cardiomyopathy, and valvular heart disease [3]. Because asymptomatic structural and functional abnormalities are important HF precursors that can be identified with non-invasive imaging, it has been recommended that imaging should play a central role in HF diagnosis and evaluation [1,2]. In this regard, echocardiography and magnetic resonance imaging (MRI) are two modalities that offer the most [4]. In comparison to echocardiography that has been routinely used in the practice of clinical cardiology, cardiac MRI is relatively new. However, owing to its superior spatial resolution and soft-tissue contrast, it has grown rapidly and become a clinical standard for volumetric quantification of systolic function in the left ventricle (LV) [5]. Especially, cardiac MRI offers high reproducibility for measuring LV ejection fraction (EF) that plays a critical role in HF classification [6]. A disadvantage of cardiac MRI over echocardiography in HF evaluation is that volumetric indices including EF are not responsive to diastolic dysfunction. It has been suggested that echocardiography and cardiac MRI should play complementary roles in HF diagnosis and evaluation [4].

The presented work aims to investigate the potential of cardiac MRI for evaluating both systolic and diastolic dysfunction of the LV in HF patients. This is clinically desirable because HF evaluation with a single imaging test would improve diagnostic reproducibility and increase patient throughput. In addition, HF has become a common cause of hospitalization, especially in aged patients. By improving HF diagnosis and management, our work may lower hospitalization rates and healthcare costs [7]. Furthermore, cardiac MRI would be the only choice of image modality in HF patients with a poor echocardiographic window [4]. Physiologically, LV functions like a pump that translates myocardial force into systolic and diastolic motion for generating cardiac output. It is thus logical that LV function may be assessed by imaging and measuring myocardial walls during systolic contraction and diastolic relaxation. In this work, we sought to improve HF evaluation by analyzing LV wall motion with real-time MRI and feature tracking.

Feature tracking is a technique similar to speckle tracking imaging in echocardiography [8]. This technique can track the spatial position of LV wall boundaries along the time in cardiac images and has been demonstrated to be effective for myocardial strain or strain rate analysis [9,10,11]. In most studies on strain analysis, a conventional retrospective cine technique [12] has been used to acquire cardiac images for feature tracking. This technique collects data from several different cardiac cycles and retrospectively reconstructs them into a set of multi-phase images within a single virtual cardiac cycle. Due to multi-cycle data combination, retrospective cine permits the visualization of LV wall motion consistent over different cardiac cycles within the data acquisition window, but does not provide information about short-lived motion that may vary from one cardiac cycle to another. In contrast, real-time imaging, which has been demonstrated to be feasible in recent cardiac MRI studies [13,14,15], offers the ability to visualize both long-lasting and short-lived motion for improved quantitative motion analysis.

The presented work was therefore to validate a real-time cardiac MRI approach to feature tracking for quantitative assessment of LV wall motion. We expected that LV wall motion analysis with real-time feature tracking would provide diagnostic information complementary to volumetric measurements for HF evaluation. To demonstrate this approach, we conducted a cardiac MRI study with healthy volunteers and HF patients. Quantitative assessment of LV function was performed using real-time LV wall motion analysis in combination with volumetric measurements. We investigated whether this combination would offer the ability to evaluate both systolic and diastolic dysfunction in HF patients.

## 2. Materials and Methods

In a cardiac MRI study, we collected real-time short-axis images over a series of sequential cardiac cycles from healthy volunteers and HF patients. Feature tracking was used to measure LV wall motion velocity along the radial and circumferential directions in real-time images. This study was to demonstrate that real-time motion velocity measurements would provide LV function assessment complementary to volumetric measurements and offer the potential for improved HF evaluation with cardiac MRI.

### 2.1. Experimental Study

The experimental MRI study was to collect both retrospective cine and real-time cardiac images on the short-axis plane in healthy volunteers and HF patients. The retrospective cine was used as a reference for real-time imaging in LV wall motion analysis with feature tracking.

#### 2.1.1. Study Design

This study was approved by the St. Francis Hospital Institutional Review Board. Written informed consent was signed by each individual participant. We recruited healthy volunteers and HF patients. The study exclusions included metallic hazards, pacemaker/defibrillator device and claustrophobia. In addition, the subjects with a resting heart rate greater than 90 bpm or a systolic blood pressure greater than 180 mmHg were excluded. A healthy volunteer was recruited if there was no history of cardiovascular disease and both ECG and echocardiography examinations were normal. A HF patient was recruited if he/she was hospitalized with diagnosed HF during the last six months.

Based on LV EF, we divided the recruited HF patients into two groups, the patients with EF lower than normal (i.e., EF < 50%) and those with preserved EF (i.e., EF ≥ 50%). In cardiology guidelines [1,2], HF with preserved EF is termed as HFpEF, and the HF patients with EF < 50% may be further divided into two subgroups, one with reduced EF (<40%) and the other with mid-range EF (40–49%). In this study, the subgroups with reduced and mid-range EF were combined together because they all had systolic dysfunction (severe or minor) that could be evaluated with volumetric measurements. In contrast, the patients with HFpEF, as typically associated with diastolic dysfunction, could not be evaluated adequately with volumetric measurements. Our study goal was to find a new cardiac MRI index that would provide HF evaluation complimentary to volumetric measurements, i.e., that would offer the ability to differentiate the patients with HFpEF from the healthy volunteers.

#### 2.1.2. Data Acquisition

Every subject was scanned with both retrospective cine and real-time imaging. All the cardiac MRI scans were run using a balanced steady state free precession (bSSFP) sequence with a 12-channel coil array on a 1.5T clinical scanner (Siemens Healthineers, Erlangen, Germany).

In each cardiac MRI session, a retrospective cine scan was first run with breath-holding and ECG-gating [16]. The retrospective cine data were collected with the following acquisition parameters: FOV 340 × (220–250) mm, voxel 1.5–1.9 mm, segments 5–8, iPAT factor 2, ECG-synchronized phases 30, TR/TE 2.6/1.3 ms, FA 50°–75°, short-axis slices 10, slice thickness 8 mm, slice gap 2 mm, and bandwidth 1420 Hz.

Following the retrospective cine scan, a real-time imaging scan was run during free-breathing and without ECG-gating. The real-time imaging data were collected using radial sampling with the following acquisition parameters: FOV 230–250 mm, voxel 1.5–1.9 mm, TR/TE 2.2–3.0/1.1–1.5 ms, FA 50°–75°, short-axis slices 10, slice thickness 8 mm, slice gap 2 mm, bandwidth 1510 Hz, and radial views 3072.

#### 2.1.3. Image Reconstruction

The retrospective cine images were reconstructed online using a software tool provided by the MRI manufacturer. Each series of retrospective cine images included 30 phases in a single cardiac cycle with a nominal temporal resolution of 30–40 ms. The real-time images were reconstructed offline in MATLAB (The MathWorks, Inc., Natick, MA, USA) with a lab-developed reconstruction algorithm based on published real-time imaging studies [13,17]. Each series of real-time images included 7–11 cardiac cycles with a temporal resolution of 18–30 ms. There were 33–48 phases in every cardiac cycle within the data acquisition window.

### 2.2. Image Analysis and Measurements

Image segmentation and feature tracking were performed for identifying and tracking endocardial borders in the LV. The resultant border contours and temporal trajectories were used to measure LV volumes and LV wall motion velocity in retrospective cine and real-time images.

#### 2.2.1. Volumetric Measurements with Retrospective Cine and Real-Time Images

Volumetric measurements were performed in the LV with retrospective cine images by following a standard procedure recommended by the Society of Cardiovascular Magnetic Resonance [5]. A software tool dedicated for cardiac image processing (CVI42, Circle Cardiovascular Imaging Inc., Calgary, AB, Canada) was used to delineate the endocardial borders and calculate the LV blood pool areas at the end of diastole and systole in each image slice. The measurements from all the short-axis slices were used to calculate LV volumetric indices, including end-diastolic volume (EDV), end-systolic volume (ESV), stroke volume (SV) and ejection fraction (EF), as in previous studies [18,19,20].

A cardiac cycle with the best image quality was manually selected from the real-time images in each image slice. This cycle was processed with the same procedure as in volumetric measurements with retrospective cine images: The software tool CVI42 was used to delineate the endocardial borders and calculate the LV blood pool areas at the end of diastole and systole. The measurements from all the short-axis slices were used to generate real-time LV volumetric indices in each subject.

#### 2.2.2. Velocity Measurements with Retrospective Cine and Real-Time Images

Velocity measurements were based on an image segmentation and a feature tracking algorithm from published studies [10,21]. As available software tools could not process real-time images collected over multiple cardiac cycles, we implemented both algorithms with MATLAB. The image segmentation algorithm [21] was automatic which provided the delineation of LV endocardial borders based on both image contrast between the blood pool and the myocardium and prior information about ventricular geometry and structure. The segmentation results were manually reviewed and corrected if necessary. The feature tracking algorithm [10] was to track the motion of LV endocardial borders identified with image segmentation. As illustrated in Figure 1, this algorithm first interpolated a set of “line cuts” positioned radially across the endocardial border and separated uniformly along the border in every image time frame. We then extracted the “line cuts” around each voxel along the endocardial border and aligned them into a feature image for that voxel. The voxel-wise tracking was performed by searching for the feature images which geometrically matched best in neighboring time frames [22]. The radial displacement was calculated by the radius change of each voxel between two neighboring image time frames and the circumferential displacement by the angular change. The radial and circumferential velocity measurements were given by the radial and circumferential displacements divided by the time interval between neighboring time frames.

#### 2.2.3. LV Wall Motion Analysis for HF Evaluation

The feature tracking (Figure 1) gave a time series of LV wall motion velocity measurements along the radial and circumferential directions at every spatial location of the LV endocardial border. Like in previous studies [23], the peak velocity during the diastolic phase was measured within each cardiac cycle, respectively, along the radial and along the circumferential direction. In each subject, the peak velocity measurements were averaged over all the slices. In real-time imaging, they were also averaged over all the cardiac cycles. The averaged peak velocity was used in statistical analysis.

A Pearson correlation coefficient [24] was used to measure the temporal correlation between radial and circumferential velocity measurements in each slice. It should be noted that, due to LV twisting, circumferential motion was clockwise in the basal slices and counter-clockwise in the apical slices. As a result, the correlation coefficient between radial and circumferential velocity measurements was positive in the basal slices and negative in the apical slices. A cardiac MRI index, torsion correlation, was calculated by the difference between the averaged correlation coefficient over all the basal slices with clockwise rotation and that over all the apical slices with counter-clockwise rotation. We investigated whether the HF patients with EF < 50% and the patients with HFpEF would be differentiated from the healthy volunteers in a scatter plot of EF against torsion correlation.

### 2.3. Statistical Analysis

The measurements of EDV, ESV, SV, EF, peak velocity and torsion correlation from both retrospective cine and real-time images were compared among the healthy volunteers, the HF patients with EF < 50%, and the patients with HFpEF, using a one-way ANOVA method [25]. A Pearson correlation coefficient [24] was calculated to evaluate the correlation between EF and peak velocity. In each statistical analysis, an Anderson-Darling test [26] was used to assess data normality. The statistical results were presented with box plots that provided the median, 25th and 75th percentiles of the index measurements. The data summary was shown as mean ± standard deviation. A P value of less than 0.05 was considered to be statistically significant. All the statistics were carried out in MATLAB.

## 3. Results

### 3.1. Patient Characteristics

A total of 16 healthy volunteers and 18 HF patients were recruited. Among the HF patients, ten had preserved EF and the other eight had EF < 50%. Table 1 and Table 2 provide the New York Heart Association (NYHA) classification of heart failure, a list of the relevant cardiovascular and pulmonary problems and the cardiac MRI assessment, respectively, in every HF patient with EF < 50% and in every patient with HFpEF. The cardiac MRI assessment indicated that LV volumetric measurements effectively detected systolic dysfunction in HF with EF < 50% while they could not differentiate the HFpEF patients from the healthy volunteers.

### 3.2. Collected Images and Volumetric Measurements

Figure 2A provides examples of the retrospective cine and real-time images collected, respectively, in a healthy volunteer and in a HF patient. In comparison, retrospective cine gave better image quality because every image was generated from the retrospectively-combined data in several different cardiac cycles. Real-time images were more informative about short-lived motion that varied from one cardiac cycle to another because the entire time-series images permitted the real-time visualization of a series of sequential cardiac cycles with a temporal resolution of 18–30 ms.

Figure 2B provides the one-way ANOVA statistics for the volumetric measurements with both retrospective cine and real-time images. The two measurements were comparable: The HF patients with EF < 50% gave a larger EDV and ESV, and a lower EF, than the healthy volunteers (*p* ≤ 0.002), while two groups did not present a significant difference in SV (*p* ≥ 0.8). No significant difference was found between the healthy volunteers and the patients with HFpEF in most volumetric indices (*p* ≥ 0.05) except that real-time imaging detected a difference in EDV (*p* = 0.006) and SV (*p* = 0.005). These measurements were consistent with the clinical evaluation in Table 1 and Table 2. The HF patients with EF < 50% and those with preserved EF presented a significant difference in ESV, SV and EF (*p* ≤ 0.02) while no difference was found in EDV (*p* = 0.2).

### 3.3. Velocity Measurements with Feature Tracking

Figure 3A gives examples of the LV border motion trajectories identified with feature tracking in both retrospective cine and real-time images from a healthy volunteer. Figure 3B provides the corresponding velocity measurements from the LV border trajectories. During systole, LV walls presented inward motion along the radial direction. At the same time, there was clockwise rotation in the basal slice and counter-clockwise rotation in the apical slice. During diastole, the motion reversed both radially and circumferentially. In both retrospective cine and real-time images, the radial and circumferential velocity measurements were found to be in-phase in the basal slice while they were out-of-phase in the apical slice, indicating that there was a twisting action with radial contraction and an untwisting action with radial relaxation. However, the retrospective feature tracking provided motion trajectories only within a single cardiac cycle. These single-cycle trajectories showed long-lasting motion consistent over different cardiac cycles. In contrast, real-time feature tracking generated motion trajectories over a series of sequential cardiac cycles. These multi-cycle trajectories provided information not only about long-lasting motion but also about short-lived motion that varied from one cardiac cycle to another. Correspondingly, the velocity measurements with real-time imaging were more informative than those with retrospective cine because they showed the variation of radial and circumferential velocity not only within every single cardiac cycle but also across different cardiac cycles.

### 3.4. Ventricular Function Assessment

Figure 4A provides the one-way ANOVA statistics for peak velocity measurements in the healthy volunteers, HF patients with EF < 50%, and patients with HFpEF. The statistics were found to be similar to those for EF measurements (Figure 2B) in both retrospective cine and real-time images: The healthy volunteers and HFpEF patients gave higher peak velocity measurements along the radial and circumferential directions than those in the HF patients with EF < 50% (*p* ≤ 0.02). No significant difference was found between the healthy volunteers and the patients with HFpEF (*p* ≥ 0.5). The correlation analysis (Figure 4B) indicated that the peak velocity and EF measurements were strongly correlated (R ≥ 0.55) in both retrospective cine and real-time images.

Figure 5 provides examples of the correlation analysis between radial and circumferential velocity measurements, respectively, in a healthy volunteer, in a HF patient with EF < 50%, and in a HF patient with preserved EF. It was found that the correlation between radial and circumferential velocity measurements was positive in the basal slices with clockwise rotation and negative in the apical slices with counter-clockwise rotation. The correlation calculated from retrospective cine images was not considerably different among the three subjects. The resultant torsion correlation was also comparable. In contrast, the correlation from real-time images was considerably different among the three subjects. The resultant torsion correlation was the highest in the healthy volunteer, and the lowest in the HFpEF patient.

Figure 6A gives the one-way ANOVA statistics for the measurements of torsion correlation in the healthy volunteers, HF patients with EF < 50% and patients with HFpEF. Significant differences were found in the real-time measurements (*p* ≤ 0.01), but not in the retrospective measurements (*p* ≥ 0.5). Figure 6B shows the scatter plots of EF against torsion correlation, respectively from retrospective cine and from real-time imaging. The HF patients with EF < 50% were separated from the other two groups due to the difference of EF in both plots. The healthy volunteers and the patients with HFpEF were mixed in the plots from retrospective cine images while they were differentiated in those from real-time images.

## 4. Discussion

### 4.1. Main Findings

This study introduced a real-time cardiac MRI approach to feature tracking for measuring LV wall motion velocity along the radial and circumferential directions on the short-axis plane. A cardiac MRI index, torsion correlation was calculated from correlation analysis between the radial and circumferential velocity measurements for LV function assessment. We investigated torsion correlation in 16 healthy volunteers and 18 HF patients (8 with EF < 50% and 10 with preserved EF), and found the following:(1)Torsion correlation was statistically different between the healthy volunteers and the patients with HFpEF (one-way ANOVA, *p* < 0.001) while volumetric indices were comparable, indicating correlation analysis provided diagnostic information complementary to volumetric measurements.(2)In the scatter plots of EF against torsion correlation, the HF patients with EF < 50%, the patients with HFpEF, and the healthy volunteers were well differentiated, indicating the potential of correlation analysis for improved HF evaluation.

### 4.2. Rationales for Correlation Analysis between Radial and Circumferential Motion

The contractile shortening of myocardial fibres is at most 15–20% while EF can reach >50% in a normal ventricle. It has been realized in previous works [27] that this discrepancy should be attributed to the involvement of twisting. Logically, there have been a number of cardiac MRI studies on LV twisting mechanics for quantitative assessment of ventricular function. Several MRI techniques, including grid tagging, strain-encoded imaging, feature tracking and velocity-encoded imaging [11,28,29,30], have been used to measure myocardial strain, peak velocity or relaxation time constant during LV twisting [8,23,31,32,33,34]. However, most of these studies have sought to analyse radial and circumferential motion separately instead of the relationship between two motion components. It should be noted that, although both radial and circumferential motion components are associated with oblique fibre orientation, they behave differently: Circumferential rotation is strongest in the slices close to the apex [35] while radial motion in those between the base and the mid-ventricle [23]. This implies that whether the radial and circumferential motion components would coordinate with each other along the time should play a role in LV contraction and relaxation. In a previous study [36], it has been reported that diastolic dysfunction would cause increased LV stiffness. As stiffness is an important parameter that determines the time constant of a mechanic system like LV, it is logical that the time synchronization of LV wall motion may be compromised in HFpEF patients. Accordingly, we expect that the HFpEF patients should present more temporal inconsistency between the radial and the circumferential motion components than that in the healthy volunteers.

A contribution of the presented work was the introduction of correlation analysis between the radial and the circumferential motion in LV walls. This correlation analysis provided a new cardiac MRI index, torsion correlation, for assessing how well LV torsional and radial motion would coordinate with each other along the time over a series of sequential cardiac cycles. We demonstrated that torsion correlation was different among the healthy volunteers, the HF patients with EF < 50% and the patients with HFpEF (Figure 6). In contrast, like EF, the measurements of radial and circumferential peak velocity [23,33] could not differentiate between the healthy volunteers and the patients with HFpEF (Figure 4). This implied that separate analysis of radial and circumferential motion was not an ideal approach to extracting diagnostic information complementary to volumetric measurements. Instead, the relationship between radial and circumferential motion should be valuable for evaluating systolic and diastolic dysfunction in the LV. The significance of our work would therefore be the provision of a new direction to seek new cardiac MRI indices for improved HF evaluation.

### 4.3. Retrospective Cine vs. Real-Time Imaging

There is a technical reason why no extensive investigation has been made on the relationship between radial and circumferential motion. Since its inception, cardiac MRI has been largely dependent on retrospective cine. Because retrospective cine uses the data collected from several different cardiac cycles to reconstruct multi-phase images within a single virtual cardiac cycle, it permits the visualization of only long-lasting motion consistent across different cardiac cycles within the data acquisition window. Due to lack of the information about short-lived motion that may vary from one cardiac cycle to another, retrospective cine images cannot provide sufficient temporal details about LV wall motion for correlation analysis between different motion components. As demonstrated in Figure 6, we found that the motion correlation from retrospective cine images gave no significant differences between the healthy volunteers and the patients with HFpEF.

To analyse short-lived motion that would be important to correlation analysis between radial and circumferential motion, we introduced real-time imaging in the presented work. Although real-time imaging gave worse image quality than that in retrospective cine (Figure 2A), it allowed visualization of a series of sequential cardiac cycles in real-time and provided evaluation of the short-lived motion in every different cardiac cycle. By correlation analysis over a sufficiently long time (7–11 cardiac cycles in the presented work), torsion correlation was found to be effective for assessing the difference in ventricular function among the healthy volunteers, the HF patients with EF < 50% and the patients with HFpEF.

### 4.4. Study Limitation

The presented work is a proof-of-concept study on real-time MRI feature tracking for quantitative assessment of LV wall motion in HF patients. As HF etiology is complicated [1,2,3], a small number of HF patients are not enough for evaluate the full potential of real-time LV wall motion analysis. However, this study indicates that real-time LV wall motion analysis can differentiate HFpEF patients from healthy subjects. Therefore, the significance of a further study on real-time MRI feature tracking with a large number of HF patients would be high.

### 4.5. Clinical Signficiance

HF evaluation and management is complicated and challenging, leading to a high hospitalization cost for health economies worldwide [3,7]. The presented work has the potential to enable a non-invasive approach to assessing both systolic and diastolic dysfunction in HF patients with a single cardiac MRI test. This may improve in-hospital HF management and reduce the cost of HF hospitalization.

## 5. Conclusions

Real-time MRI feature tracking enables an approach to correlation analysis between the radial and the circumferential velocity measurements of LV wall motion over a series of sequential cardiac cycles. This approach provides a cardiac MRI index, torsion correlation, complementary to EF, for ventricular function assessment, and offers the potential for improved HF evaluation.

## Figures and Tables

**Figure 1 diagnostics-12-02946-f001:**
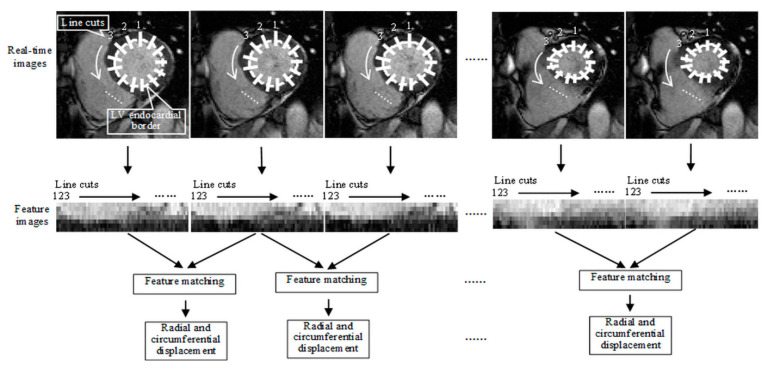
Illustration of feature tracking algorithm for voxel-wise measurements of LV endocardial border motion along the radial and circumferential directions. In each time frame, a set of “line cuts” normal to the LV endocardia border were interpolated from the Cartesian image voxels and arranged to form a feature image for each voxel along the LV endocardial border. The voxel-wise tracking across time frames was performed by searching for the feature images which geometrically matched best. The radial displacement was given by the radius change of each voxel between two neighboring image time frames and the circumferential displacement by the angular change.

**Figure 2 diagnostics-12-02946-f002:**
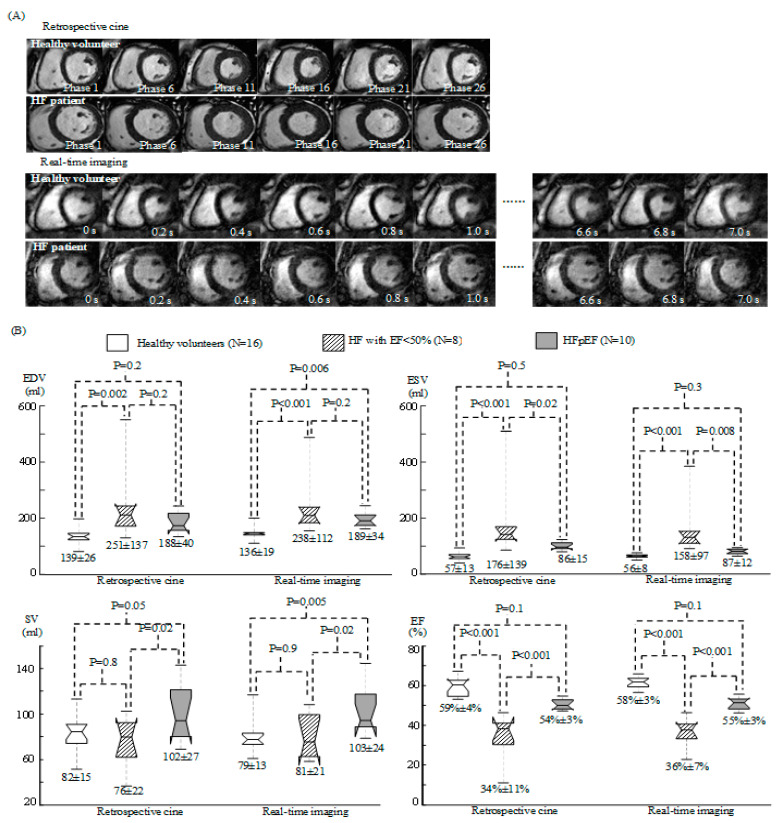
(**A**) Selected time frames of the retrospective cine and real-time images in a healthy volunteer and a heart failure patient. (**B**) Box plots for EDV, ESV, SV and EF measurements with retrospective cine and real-time imaging in the healthy volunteers and HF patients. The numbers above the boxes provide the P values from one-way ANOVA statistics and those below the boxes are the mean and standard deviation in each subject group.

**Figure 3 diagnostics-12-02946-f003:**
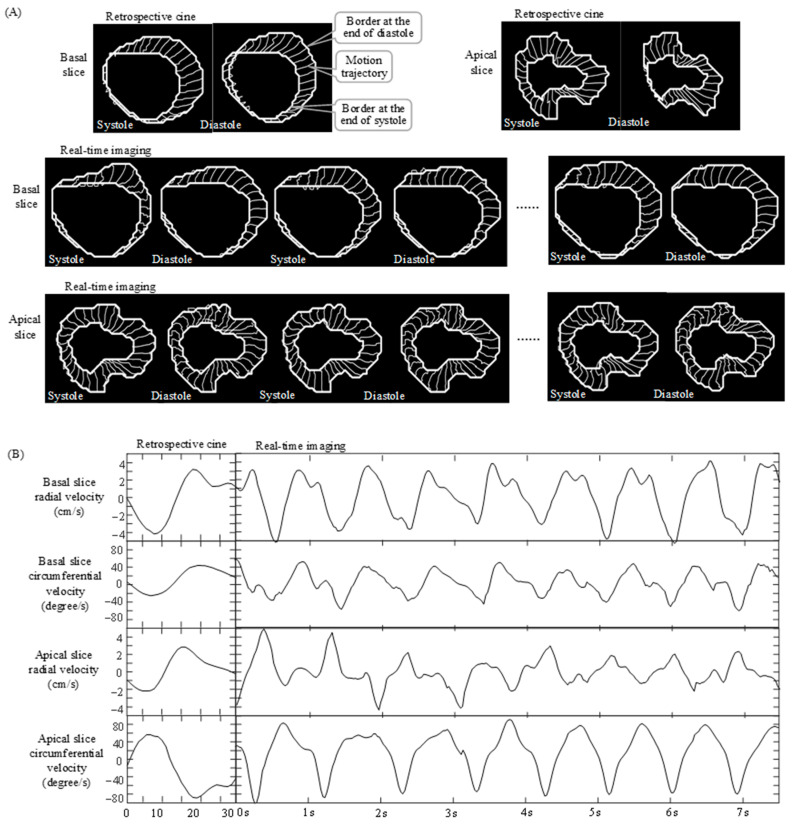
Velocity measurements along the radial and circumferential directions in retrospective cine and real-time images from a healthy volunteer. (**A**) Motion trajectories of selected voxels along LV endocardial borders during systole and diastole within each cardiac cycle, respectively, in a basal and in an apical slice. (**B**) Radial and circumferential velocity measurements from the LV border trajectories in (**A**). The radial and circumferential velocity components were in-phase in the basal slice, but out-of-phase in the apical slice. Real-time measurements showed short-lived motion that varied from one cardiac cycle to another while retrospective measurements showed only long-lasting motion consistent over different cardiac cycles.

**Figure 4 diagnostics-12-02946-f004:**
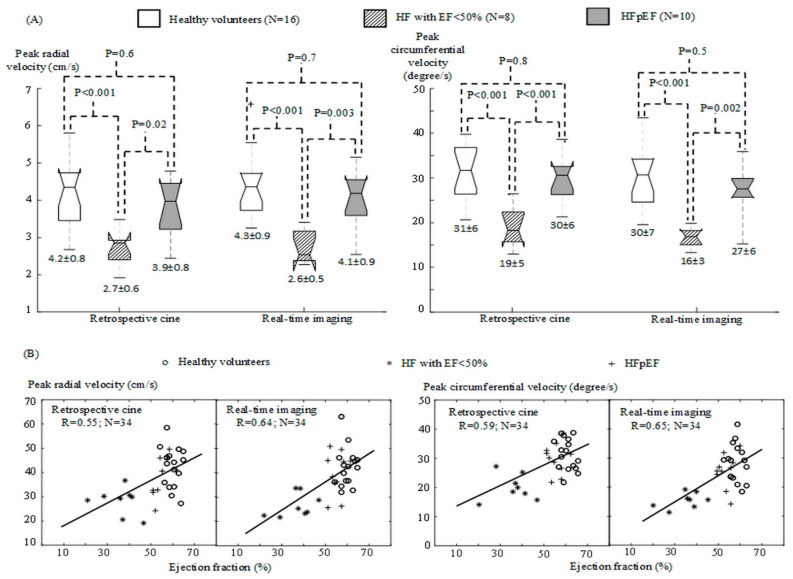
Peak velocity measurements with retrospective cine and real-time imaging in the healthy volunteers and HF patients. (**A**) Box plots for peak radial and circumferential velocity measurements in retrospective cine and real-time images. The numbers above the boxes provide *p* values from one-way ANOVA statistics and those below the boxes are the mean and standard deviation in each subject group. (**B**) Pearson correlation coefficients between peak velocity measurements and EF in retrospective cine and real-time images.

**Figure 5 diagnostics-12-02946-f005:**
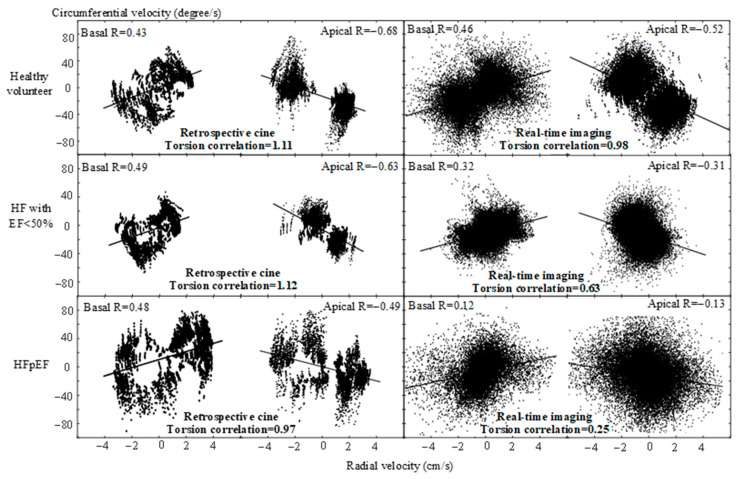
Examples of correlation analysis between radial and circumferential velocity measurements in retrospective cine and real-time images, respectively, from a healthy volunteer, from a HF patient with EF < 50%, and from a patient with HFpEF. The torsion correlation from real-time imaging was considerably different among three subjects while that from retrospective cine comparable.

**Figure 6 diagnostics-12-02946-f006:**
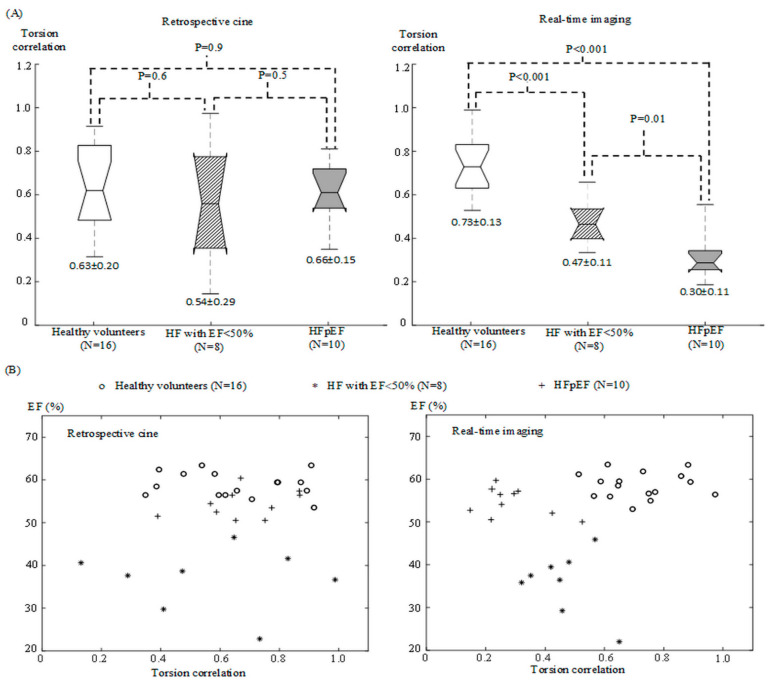
Correlation analysis with retrospective cine and real-time images from 16 healthy volunteers, and 18 HF patients. (**A**) Box plots for the measurements of torsion correlation in the healthy volunteers (N = 16), HF patients with EF < 50% (N = 8) and patients with HFpEF (N = 10). The numbers above the boxes provide P values from one-way ANOVA statistics and those below the boxes are the mean and standard deviation in each subject group. (**B**) Scatter plots of EF against torsion correlation from retrospective cine and real-time images. The patients with HFpEF and the healthy volunteers were differentiated in the plots from real-time images while they were mixed in those from retrospective cine images.

**Table 1 diagnostics-12-02946-t001:** NYHA classification, diagnosed diseases and cardiac MRI assessment of the HF patients with EF < 50%.

ID	NYHA Class of HF	Relevant Cardiovascular and Pulmonary Problems	Cardiac MRI Assessment			
			LV		RV	
			Size	Systolic Function	Size	Systolic Function
1	II	Nonischemic cardiomyopathy, shortness of breath, palpitation	Severely dilated, EDV index = 154 mL/m^2^	Moderately reduced, EF = 37%	Normal, EDV index = 90 mL/m^2^	Normal, EF = 56%
2	II	Hypertension, diabetes mellitus, thyroid nodule	Moderately dilated, EDV index = 110 mL/m^2^	Moderately reduced EF = 42%	Normal, EDV index = 92 mL/m^2^	Normal, EF = 57%
3	II	Hypertension, cardiomyopathy, chronic obstructive pulmonary disease, internal cardiac defibrillator procedure, hyperkalemia, dizziness	Normal, EDV index = 73 mL/m^2^	Severely reduced, EF = 27%	Normal, EDV index = 72 mL/m^2^	Moderately reduced, EF = 37%
4	II	Atrial fibrillation, stroke, nonischemic cardiomyopathy, transient ischemic attack, sleep apnea, chest pain, palpitation	Normal, EDV index = 90 mL/m^2^	Severely reduced, EF = 33%	Normal, EDV index = 76 mL/m^2^	Mildly reduced, EF = 41%
5	II	Atrial fibrillation, nonischemic cardiomyopathy, transient ischemic attack, chest pain, palpitation	Normal, EDV index = 90 mL/m^2^	Moderately reduced, EF = 39%	Normal, EDV index = 90 mL/m^2^	Mildly reduced, EF = 40%
6	II	Shortness of breath	Severely dilated, EDV index = 293 mL/m^2^	Severely reduced, EF = 10%	Normal, EDV index = 90 mL/m^2^	Moderately reduced, EF = 32%
7	II	Restrictive cardiomyopathy, dyspnea on exertion	Moderately dilated, EDV index = 114 mL/m^2^	Moderately reduced, EF = 40%	Normal, EDV index = 67 mL/m^2^	Normal, EF = 56%
8	II	Ventricular tachycardia, premature ventricular contraction, first degree atrioventricular block, non-sustained ventricular tachycardia, first degree heart block, left anterior fascicular block, hypertension, coronary artery disease, chest pain	Normal, EDV index = 74 mL/m^2^	Mildly reduced, EF = 46%	Normal, EDV index = 55 mL/m^2^	Normal, EF = 51%

**Table 2 diagnostics-12-02946-t002:** NYHA classification, diagnosed diseases and cardiac MRI assessment of the patients with HFpEF.

ID	NYHA Class of HF	Relevant Cardiovascular and Pulmonary Problems	Cardiac MRI Assessment			
			LV		RV	
			Size	Systolic Function	Size	Systolic Function
1	II	Pulmonary hypertension, cardiomyopathy, dilated cardiomyopathy, pulmonary embolism, atherosclerosis of arteries of extremities, chest pain,	Normal, EDV index = 75 mL/m^2^	Normal, EF = 53%	Normal, EDV index = 79 mL/m^2^	Normal, EF = 51%
2	II	Pulmonary hypertension, cardiomyopathy, dilated cardiomyopathy	Mildly dilated, EDV index = 111 mL/m^2^	Normal EF = 59%	Mildly dilated, EDV index = 113 mL/m^2^	Normal, EF = 59%
3	III	Hypertension, pulmonary hypertension, atrial fibrillation, obstructive sleep apnea	Mildly dilated, EDV index = 104 mL/m^2^	Normal EF = 57%	Severely dilated, EDV index = 141 mL/m^2^	Mildly reduced, EF = 42%
4	I	Cardiomyopathy, paroxysmal atrial fibrillation	Normal, EDV index = 72 mL/m^2^	Normal, EF = 50%	Normal, EDV index = 66 mL/m^2^	Low normal, EF = 49%
5	I-II	Nonischemic cardiomyopathy, atrial fibrillation	Normal, EDV index = 87 mL/m^2^	Normal, EF = 54%	Normal, EDV index = 88 mL/m^2^	Normal, EF = 50%
6	I-II	Nonischemic cardiomyopathy, atrial fibrillation	Normal, EDV index = 93 mL/m^2^	Normal, 54%	Normal, EDV index = 87 mL/m^2^	Normal, EF = 55%
7	I-II	Nonischemic cardiomyopathy, atrial fibrillation	Mildly dilated, EDV index = 101 mL/m^2^	Normal, EF = 59%	Mildly dilated, EDV index = 104 mL/m^2^	Normal, EF = 63%
8	III	Arteriosclerotic heart disease, pulmonary embolism on right, obstructive sleep apnea, hyperlipidemia, morbid obesity	Normal, EDV index = 86 mL/m^2^	Normal, EF = 50%	Moderately dilated, EDV index = 120 mL/m^2^	Mildly reduced, EF = 41%
9	III	Hypertension, pulmonary emboli, chronic obstructive pulmonary disease, asthma-COPD overlap syndrome, nonischemic cardiomyopathy, atrial fibrillation	Normal, EDV index = 66 mL/m^2^	Normal, EF = 52%	Normal, EDV index = 69 mL/m^2^	Normal, EF = 52%
10	II	Hypertension, coronary artery disease, non-ST elevated myocardial infarction, ventricular fibrillation, ventricular tachycardia, ischemic cardiomyopathy, ischemic heart disease, acute respiratory failure with hypoxia, hyperlipidemia, shortness of breath	Normal, EDV index = 95 mL/m^2^	Normal, EF = 51%	Normal, EDV index = 80 mL/m^2^	Normal, EF = 62%

## Data Availability

The datasets generated and/or analyzed during this study are available upon request.

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
