# Peer review of "Left Ventricle Wall Motion Analysis with Real-Time MRI Feature Tracking in Heart Failure Patients: A Pilot Study"

_diagnostics, 2022, doi:10.3390/diagnostics12122946_

Round 1

Reviewer 1 Report

In manuscript titled “Quantitative Assessment of Left Ventricle Wall Motion with Real-time MRI Feature Tracking in Heart Failure Patients”, the authors aim to validate a new cardiac MRI index, matching LV wall motion analysis in combination with volumetric measurement. 

The manuscript il well written and conclusions are potentially interesting in the field, moreover I have some comments:

1) the main limitation of this study is the small number of patients involved, nevertheless the results are interesting. I think authors should rephrase the title as follow: “Quantitative Assessment of Left Ventricle Wall Motion with Real-time MRI Feature Tracking in Heart Failure: a pilot study”.

2) author conclude that real-time MRI provide a cardiac MRI index, complementary to EF, for quantitative assessment of ventricular function. This improved HF evaluation, helping in differentiate HFpEF patients from healthy subjects. Could authors briefly explain what is the clinical application of this finding in the real life? i.e. reduction of hospitalizations? reduction of hectare costs? 

3) do the patients enrolled signed informed consent?

4) could the method presented reduce hospitalizations? I find this could be a pivotal result, also considering the high costs of HF hospitalization, particularly in aged patients. Please, briefly discuss this point also mentioning the following paper “Tersalvi G et al. Acute heart failure in elderly patients: a review of invasive and non-invasive management. J Geriatr Cardiol. 2021 Jul 28;18(7):560-576. doi: 10.11909/j.issn.1671-5411.2021.07.004.”

Reviewer 2 Report

All measurements described are semiquantitative. Real-time measurements cannot be taken as relevant without an ECG signal. The measurement errors increases with the calculation of the indices.

Round 2

Reviewer 1 Report

I appreciate authors efforts in revise the manuscript that results improved.

Reviewer 2 Report

This manuscript is tasked with increasing the importance and role of MRI techniques in the diagnosis and differentiation of patients with different categories of heart failure; for a group of patients in whom echocardiography for some reason it is not possible to do. Since Doppler echocardiography is the gold standard esspecially for the patients with HFpEF and HFmEF, it would be of interest to compare the hemodynamic and mechanic LV indices of both imaging techniques, in order to define more preciselly the diagnostic algorithms.